# Comparative Proteomics of Root Apex and Root Elongation Zones Provides Insights into Molecular Mechanisms for Drought Stress and Recovery Adjustment in Switchgrass

**DOI:** 10.3390/proteomes8010003

**Published:** 2020-02-19

**Authors:** Zhujia Ye, Sasikiran Reddy Sangireddy, Chih-Li Yu, Dafeng Hui, Kevin Howe, Tara Fish, Theodore W. Thannhauser, Suping Zhou

**Affiliations:** 1Department of Agricultural and Environmental Sciences, College of Agriculture, Tennessee State University, 3500 John Merritt Blvd, Nashville, TN 37209, USA; zhujiaye117@gmail.com (Z.Y.); sangisasi@gmail.com (S.R.S.); 2Department of Biological Sciences, Tennessee State University, 3500 John Merritt Blvd, Nashville, TN 37209, USA; e390701@gmail.com (C.-L.Y.); dhui@tnstate.edu (D.H.); 3Functional & Comparative Proteomics Center, USDA-ARS, Ithaca, NY 14853, USA; kevin.howe@ars.usda.gov (K.H.); tara.fish@usda.gov (T.F.)

**Keywords:** root-tip zone-specific quantitative proteomics, drought, cell proliferation, cell wall remodeling, hydrophilic proteins, stress proteins, phytohormones

## Abstract

Switchgrass plants were grown in a Sandwich tube system to induce gradual drought stress by withholding watering. After 29 days, the leaf photosynthetic rate decreased significantly, compared to the control plants which were watered regularly. The drought-treated plants recovered to the same leaf water content after three days of re-watering. The root tip (1cm basal fragment, designated as RT1 hereafter) and the elongation/maturation zone (the next upper 1 cm tissue, designated as RT2 hereafter) tissues were collected at the 29th day of drought stress treatment, (named SDT for severe drought treated), after one (D1W) and three days (D3W) of re-watering. The tandem mass tags mass spectrometry-based quantitative proteomics analysis was performed to identify the proteomes, and drought-induced differentially accumulated proteins (DAPs). From RT1 tissues, 6156, 7687, and 7699 proteins were quantified, and 296, 535, and 384 DAPs were identified in the SDT, D1W, and D3W samples, respectively. From RT2 tissues, 7382, 7255, and 6883 proteins were quantified, and 393, 587, and 321 proteins DAPs were identified in the SDT, D1W, and D3W samples. Between RT1 and RT2 tissues, very few DAPs overlapped at SDT, but the number of such proteins increased during the recovery phase. A large number of hydrophilic proteins and stress-responsive proteins were induced during SDT and remained at a higher level during the recovery stages. A large number of DAPs in RT1 tissues maintained the same expression pattern throughout drought treatment and the recovery phases. The DAPs in RT1 tissues were classified in cell proliferation, mitotic cell division, and chromatin modification, and those in RT2 were placed in cell wall remodeling and cell expansion processes. This study provided information pertaining to root zone-specific proteome changes during drought and recover phases, which will allow us to select proteins (genes) as better defined targets for developing drought tolerant plants. The mass spectrometry proteomics data are available via ProteomeXchange with identifier PXD017441.

## 1. Introduction

Switchgrass (*Panicum virgatum*), a C4 grass, is a leading dedicated biofuel feedstock candidate plant species in the US due to its broad adaptability, high biomass yield, rapid growth rate, high tolerance to drought condition, ability to grow in low production soils, and widespread adaptability to temperate climate [1,2]. Drought tolerance is one of the most striking physiological properties of switchgrass because of its deep root system and highly efficient C_4_ metabolic pathway [3]. However, during the early stage of growth, switchgrass seedlings are very susceptible to both periodic and long-term drought conditions since their root systems are relatively shallow, occupying only the top (0–15 cm) of the soil [4]. A field trial showed that drought significantly affected seedling growth of switchgrass in the first year, furthermore, biomass yield declines, greatly, after three consecutive years of drought [5]. Thus, developing switchgrass plants with deeper root structure during the early stages of growth are the effective avenues to enhance their drought tolerance and to ensure high yield during the subsequent years in the field.

Plants can mobilize several different responses when they encounter drought. The drought escape mechanism is achieved by shortening each developmental stage as soil moisture gradually decreases, which allows plants to complete a life cycle before soil moisture declines to a detrimentally low level. The second drought tolerance mechanism is activated to maintain tissue turgor via osmotic adjustment which allows plants to maintain growth under water stress. Finally, plants also develop the drought avoidance mechanism by growing a deeper root system which pumps water from the subsoil layer, thus maintaining an adequate water supply for above ground growth of the tolerant plants [6,7,8].

It is obvious that these different types of drought stress responses involve both below and above ground tissues of plants, however, the drought tolerance mechanism exhibited in roots is paramount in importance for maintaining normal plant growth rate under water deficit condition. Roots are the very place where plants first encounter water deficit in soil, and they would sense and respond to the stress condition [9]. Under normal watering condition, the root absorbs water and nutrients from the soil and supplies them throughout the plant body, thereby playing central roles in maintaining cellular homeostasis. When water stress is more severe, roots are affected significantly and experience several structural and functional modifications including the molecular, cellular, and phenotypic changes [10,11,12]. For example, the growth of roots is limited by the declined rates of cell division, cell expansion, and the synthesis of a new cell wall. In maize (*Zea mays*) primary roots, final cell length, the length of the root elongation zone, and overall root elongation rates are all reduced under water deficit conditions [13,14]. Xiong et al. [15] reported that the inhibition of lateral root elongation is an indication of plant response to constant drought soil condition. Development of a deeper and larger root system enables plants to adapt to water limitations since it can absorb extra water from soil, postpone the dehydration in plants and alleviate drought effects [16,17]. Thus, enhancing capacity of root water uptake can help these plants to reduce the impacts from soil drought condition [18,19].

A broad range of ‘transcriptomic and proteomic studies on drought biology have revealed alteration of gene expression and proteome profiles during drought period as well as the recovery phase from the stressed conditions [20,21,22,23,24]. Proteomics has been widely used in the investigation of structural and functional interactions of proteins at a given time point at the whole proteome level. These drought-induced proteins are related to water transport, reactive oxygen species (ROS) scavenging, carbohydrate metabolism, and cell wall modification, which ultimately contribute to growth maintenance of roots under stressful conditions [25,26,27,28]. The root tip has three main zones: a zone of cell division (cells are actively dividing), a zone of elongation (cells increase in length), and a zone of maturation. Due to these distinct biological processes, the drought-induced genes/proteins in these separate zones should vary. In this study, we have conducted analysis of drought-induced proteomic changes in the basal cell division and upper elongation/maturation zones in switchgrass during drought treatment and recovery processes. The objective is to identify proteins that confer cell-specific or universal tolerance mechanisms against drought stress.

## 2. Materials and Methods

### 2.1. Construction of a “Sandwich” Drought Treatment System

A “Sandwich” drought treatment system was constructed as described previously [23] with minor modification (Figure 1). The system is comprised of double polyvinyl chloride (PVC) pipes (an outer pipe and an inner pipe), a PVC sewer and drain coupling, and a PVC sewer and drain cap (Figure 1A–C). A fiberglass screen was placed inside the inner pipe, which assisted when pulling out the plants for checking root length (Figure 1F). The “sandwich” treatment system is divided into three layers: garden soil (20.32 cm), perlite (38.1 cm), and garden soil (12.7 cm) (Figure 1D,E). WATERMARK soil water tension sensors (WATERMARK Soil Moisture Sensor, MODEL 200SS, IRROMETER Company, Inc., Riverside, CA, USA) were placed at the bottom of the top soil layer following the manufacturer’s instructions (Figure 1E) to monitor water content.

During the drought treatment period, the top-layer soil became drier gradually because the middle perlite layer would cut-off moisture movement upwards, and the moist bottom soil layer served a purpose to induce root growth downward. The system design is based on the assumption that for tolerant plants, the roots would maintain the growth rate during a gradual drying process of soil. Plants, by developing deeper roots to reach moisture in deep soil, will be able to maintain water status during the drought period.

### 2.2. Preparation of Plant Materials

Switchgrass ‘Alamo’ seeds were surface disinfected in 50% household bleach followed by three rinses in deionized water. Seeds were germinated in Magenta boxes partially filled with water and placed in an incubator shaker (50 rpm) at 25 °C for three days. Germinating seeds bearing 1-cm long radicals were transferred into seed cubes (Oasis Growing Medium, Smithers-Oasis Company, OH, USA). These seedlings were watered every three-days until they had grown to the three-leaf stage. At that stage, they were transplanted into the “sandwich” drought treatment system and maintained in an open-roof greenhouse where the temperature was set at 25–28 °C. The moisture content of the growing medium was maintained at a constant level by watering regularly (water tension < 10 centibar) until seedling roots reached the perlite layer.

Two weeks after transplanting, the root length was evaluated every three days. Three plants of each replicate group were selected, randomly, in each inspection of root length. Once roots reached the perlite layer, drought treatment was initiated by withholding water to these test plants until the soil water tension at the bottom of the top layer reached 150 centibar. The control groups received normal watering at the rate of 4 L of water every 3-days for each “sandwich” system. The soil water tension in drought groups was above 150 centibar which translates to severe drought conditions (http://www.irrometer.com/pdf/instructionmanuals/sensors/701Meter Manual-WEB.pdf). These plants were designated as severe drought treated (SDT).

The drought treatment plants were grown in triple sets. One set of the treatment plants was harvested as the SDT group. The remaining two sets were used to study plant responses to recovering from drought stress; these plants were re-watered daily for 1-day and 3-days (D1W and D3W) before tissue harvest.

The non-drought treated control groups were under well-watered condition, i.e., a range from 0 to 10 centibar. Three biological replicates were set up for every treatment group (control, SDT, D1W and D3W); and each single replicate was conducted in 10 tubes each growing two plants. A randomized block design was used for the layout of tubes in the greenhouse.

### 2.3. Root Tissue Harvest and Processing

The distal 1-cm root-tip (RT1) representing root cap, root meristem, and the distal end of the elongation zone were measured using a ruler, and cut with a sharp blade. The next 1cm long root tissues representing the elongation/maturation zone (RT2) were harvested separately [29]. After detached from the plants, each individual root section was immediately dropped into liquid nitrogen in a Magenta box which was placed in a Thermo-Flask™ Benchtop Liquid Nitrogen Containers (4.5 L, Cat #2124; Thermo Scientific, Waltham, MA, USA). Tissues were wrapped in aluminum-foil, frozen in liquid nitrogen and stored at –80 °C until protein extraction

### 2.4. Physiological Measurement

A LI-COR 6400 Portable-Photosynthesis-System (Li-Cor Inc., Lincoln, NE, USA) was used to measure leaf photosynthetic rate, stomatal conductance, and transpiration rate daily during the drought treatment and recovery period. Two fully expanded young leaves randomly selected from each plant were measured between 10:00 a.m. and 3:00 p.m. Light in the leaf chamber was set at 2000 photo·m^−2^ s^−1^. Water use efficiency (WUE) was calculated by WUE = leaf photosynthetic rate (Pn)/transpiration (Tr). Data analysis was performed using PROC GLM procedure of SAS software (Version: 9.3. SAS Inc. Cary, NC, USA). The effect of drought treatments was analyzed using a randomized block design analysis of variance (ANOVA). When a significant effect of drought treatment was detected, least significant difference (LSD) was used for multiple comparisons. Statistical analyses were performed using SAS (version 9.3; SAS Institute, Cary, NC, USA) [23].

### 2.5. Preparation of Protein Samples

Frozen samples were ground into a fine powder under liquid nitrogen using a Retsch Mixer Mill MM 400 (Retsch USA, Verder Scientific, Inc., Newtown, PA, USA). Protein extraction followed a previously described protocol [23]. Briefly, root tissue powder was washed sequentially in 10% trichloroacetic acid (TCA) in acetone, 80% methanol in 0.1 M ammonium acetate, and 80% acetone with centrifugation to pellet the powder after each wash. Protein was then extracted in a phenol (pH 8.0) and dense sodium dodecyl sulfate (SDS) buffer [30% sucrose, 2% SDS, 5% beta-mercaptoethanol (*v/w*) in 0.1 M Tris-HCl, pH 8.0]. After incubation at 4 °C for 2 h, the mixture was centrifuged at 16,000× *g* at 4 °C for 20 min. Protein in the upper phenol phase was precipitated in 0.1 M ammonium acetate in methanol after incubation overnight at −20 °C. After washes in methanol and then acetone, the air-dried protein pellets were wetted with a buffer of 500 mM triethylammonium bicarbonate (TEAB), 2 M urea, 0.1% SDS and a proteinase inhibitor cocktail for plant tissue (100 × dilution in the extraction buffer) (Sigma, St. Louis, MO, USA). Proteins were collected after centrifugation at 16,000× *g* at 4 °C for 10 min. Root proteins were concentrated using 5 kDa Corning’ Spin-X UF centrifugal concentrator (Sigma, St. Louis, MO, USA). Protein concentration was determined using a Bradford Assay Kit (Bio-Rad, Hercules, CA, USA).

### 2.6. Tandem Mass Tags (TMT) Labeling and Mass Spectrometry Analysis

One hundred µg of protein from each tissue sample was diluted (two times) with water to reduce urea to 1 M concentration. After reduction using tris-2-(carboxyethyl)-phosphine (TCEP), and cysteines blocked with methyl methanethiosulfonate (MMTS), proteins were digested with trypsin (sequencing grade modified trypsin, Promega, Madison, WI, USA) at 35 °C overnight. Peptides were labeled using the 6-plex TMT^®^ labeling kit (AB SCIEX, MA, USA) following the manufacturer’s instruction. For each experiment, the three control samples each were labeled with tags 126, 127, and 128, and the three treated samples with 129, 130, and 131. The six labeled peptides from the same treatment conditions were pooled together. Each multiplexed sample was loaded onto a cation exchange cartridge (AB SCIEX) to remove the unbound tags and SDS, followed by reverse-phase (RP) solid-phase extraction (Sep-Pak C18; Waters, MA, USA) for further cleaning of salts and other impurities. Peptides were eluted in 500 μL 50% (*v*/*v*) acetonitrile (ACN) containing 0.1% trifluoroacetic acid (TFA), and dried at reduced pressure using a CentiVac Concentrator (LabConco, Kansas City, MO, USA).

The labeled samples were pretreated with SCX cartridges (AB Sciex, Framingham, MA, USA) and desalted by Sep-Pak SPE for hpRP separation. The hpRP chromatography [29,30] was carried out using a Dionex UltiMate 3000 high performance LC system with a built-in microfraction collection option in its autosampler and UV detection (Thermo Fisher Scientific, Sunnyvale, CA, USA). The TMT-tagged tryptic peptides were reconstituted in buffer A (20 mM ammonium formate, pH 9.5, in water) and loaded onto an XTerra MS C18 column (3.5 μm, 2.1 × 150 mm, Waters) with 20 mM ammonium formate (NH_4_FA, pH 9.5) as buffer A and 80% ACN/20% 20 mM NH_4_FA as buffer B. The LC was performed using a gradient from 10 to 45% of buffer B in 30 min at a flow rate of 200 μL/min. Forty-eight fractions were collected at 1 min intervals. Fractions 1–4 contained no peptides based on UV absorbance at 214 nm and were not analyzed. The remaining fractions were pooled using a multiple fraction concatenation strategy [31], into a total of 22 samples. Beginning with fraction 5, every 22th fractions were combined into one sample, such as 5 + 27, 6 + 28, and 7 + 29, following the same order (sample pooling was provided in Appendix A
Appendix A). The pooled samples were dried, and each was reconstituted in 20 μL of 2% ACN/0.5% FA for nanoLC–MS/MS analysis [32,33].

For the low pH second dimension separation, dried samples were reconstituted with 15 µL of 2% ACN with 0.5% formic acid (FA). Samples were loaded to low pH RP chromatography, and the eluent from the analytical column was delivered to the LTQ-Orbitrap Elite (Thermo-Fisher Scientific, Waltham, MA, USA) via a “Plug and Play” nano ion source (CorSolutions LLC, Ithaca, NY, USA). The mass spectrometer was externally calibrated across the *m*/*z* range from 375–1800 with Ultramark 1621 for the Fourier transform (FT) mass analyzer, and individual runs were internally calibrated with the background polysiloxane ion at *m*/*z* 445.1200025 as a lock mass [34,35,36].

The Orbitrap Elite was operated in the positive ion mode with nanosource voltage set at 1.7 kV and capillary temperature at 250 °C. A parallel data-dependent acquisition (DDA) mode was used to obtain one MS survey scan with the FT mass analyzer, followed by isolation and fragmentation of the 15 most abundant, multiply-charged precursor ions with a threshold ion count higher than 50,000 in both the LTQ mass analyzer and the higher-energy collisional dissociation (HCD)-based FT mass analyzer at a resolution of 15,000 full width at half maximum (FWHM) and *m*/*z* 400. MS survey scans were acquired with resolution set at 60,000 across the survey scan range (*m*/*z* 375–1800). Dynamic exclusion was utilized with repeat count set to 1 with a 40 s repeat duration; exclusion list size was set to 500, 20 s exclusion duration, and low and high exclusion mass widths set to 1.5. Fragmentation parameters were set with isolation width at 1.5 *m*/*z*, normalized collision energy at 37%, activation Q at 0.25. Activation time for HCD analysis was 0.1 min. All data were acquired using XCalibur 2.1 (Thermo-Fisher Scientific) [36,37].

Mascot Daemon (Version 2.3.2; Matrix Science, Boston, MA) was used to combine pkl files generated from the 22 fractions associated with each pooled TMT labeled sample and to query them against switchgrass annotated database (http://www.phytozome.net/search.php?show=text&org=Org_Pvirgatum_v1.1). Furthermore, an exclusion list including the commonly observed peptides of keratin, porcine trypsin etc. was used to avoid accumulating spectra of uninformative ions. The database search criteria were: precursor mass tolerance set to 10 ppm and fragment tolerance set to 100 mmu. The proteolytic enzyme was set to trypsin and one missed cleavage was allowed. The MS/MS data were searched with TMT6plex as a fixed modification for N-terminal and lysine ε-amino groups and S-methylation of cysteine as a fixed modification of cysteine thiols. Oxidation of methionine and the deamidation of asparagine and glutamine were treated as variable modifications. To estimate the false discovery rate (FDR), we employed the target-decoy strategy as described previously [38]. All peptide matches reported herein have an E-value < 0.05. Protein identification required at least one peptide match with an E-value < 0.05. A false discovery rate (0.05) was calculated by searching the MS/MS data using a reversed decoy database. The unique peptides (1) or non-unique peptides (0) were listed in the proteomics report.

### 2.7. Protein Identification, Quantification, and Statistics Analysis

For a protein to be included in the quantitative analysis, it was required that at least two unique peptides be identified in all six biological samples. The intensities of reporter ions of constituent peptides were log_2_-transformed and subjected to t-test (general linear model procedure). The adjusted p value for each protein was obtained after false discovery rate (FDR) test of the raw p values of all the proteins [28,32,33]. The log_2_ transformed abundance ratios were then fit to a normal distribution. Two standard deviations (i.e., a 95% confidence level) of the log_2_-fold transformed protein abundance ratios (treated/control) with an adjusted FDR *p* ≤ 0.05 were used as the cut off for Al-induced differentially accumulated proteins (DAPs). The ratio of log_2_-fold value of proteins were transformed using the antilog function and reported as the drought-induced protein fold changes (fold (T/C)). Statistical analyses were performed using SAS (version 9.3; SAS Institute, Cary, NC, USA) [23,33,37].

The corresponding annotated accession number of switchgrass proteins in *Arabidopsis thaliana* (TAIR) and rice (*Orysa sativa*; Rice Genome Annotation Project) databases were used to categorize proteins into cellular processes and metabolic pathways using MapMan (version 3.5.1R2; 2015) [39]. The putative functions of the identified proteins were also discussed in the light of relevant information from literature and database searches on switchgrass, *A. thaliana*, rice and other plant species [23].

## 3. Results

### 3.1. Physiological Changes under Different Levels of Drought Condition

After 28 days of drought treatment, leaves on the drought-treated plants started to show signs of wilting and chlorosis (Figure 2B), with a significant lower photosynthetic rate, stomatal conductance, transpiration, and water use efficiency, compared to the same plants prior to the drought treatments as well as the well-watered control group (*p* < 0.01) (Table 1). After one-day of re-watering, plant leaves regained the turgor (Figure 2C), and leaf-water use efficiency was restored to the same level of the well-watered control plants (Table 1). But the photosynthetic rate, stomatal conductance, transpiration of treated plants were lower than the well-watered control plants (*p* < 0.01) (Table 1). After three-days of re-watering, the drought treated plants reverted to green leaves as well as all the physiological parameters including photosynthetic rate, stomatal conductance, transpiration, and water use efficiency (Table 1).

### 3.2. The Drought-Induced Proteins and Functional Classifications

In this study, 10,018 proteins were quantified (each with two or more unique peptides) in RT1 proteomes, with 6156, 7687, and 7699 in SDT, D1W, and D3W, respectively (Appendix A). Among the three treatment groups, 4497 overlapped proteins were identified and 609, 1125, and 1257 were only found in SDT, D1W, or D3W. In RT2 tissue proteomes, 9731 proteins were quantified, with 7382, 7255, and 6883 proteins in SDT, D1W, and D3W, respectively (Appendix A). When compared between two root sections, 23% (2274/10,018) of the quantified proteins were unique to RT1, and 1,987 quantified proteins were only identified in RT2 (Appendix A). For the identification of drought-induced differentially accumulated proteins (DAPs), the quantified proteins were filtered using the following criteria: protein fold (T/C) greater than twice the standard deviation (SD) which was obtained from the normal distribution curve of the whole proteome, and the FDR adjusted *p* < 0.05. In RT1, 296, 535, and 384 proteins passed the threshold values in SDT (fold change > 2.60 or < 0.38), D1W (fold change >2.20 or < 0.45) and D3W (fold change > 1.94 or < 0.52), respectively. In total, 780 drought-induced DAPs of RT1 proteins were identified (Appendix A). In RT2 tissue proteomes, 393, 587, and 321 drought-induced DAPs were identified in SDT (fold change >2.50 or < 0.40), D1W (fold change >2.33 or < 0.43), and D3W (fold change > 2.0 or < 0.5), respectively. In total, 1064 drought-induced DAPs were identified in RT2 (Appendix A).

The quantified proteins were clustered into functional pathways using MapMan based on the protein annotation in *A. thaliana* database. All of the six proteomes were placed into 36 functional groups, each containing nine or more proteins. The functional groups include: Glyoxylate cycle (1), S-assimilation (2), Ammonia metabolism (3), Fermentation (4), Metal binding (5), DNA repair (6), Pentose phosphate pathway (7), Carbohydrates metabolism (8), RNA synthesis (9), Major CHO metabolism (10), Minor CHO metabolism (11), Mitochondrial electron transport/ATP synthesis (12), TCA cycle (13), Glycolysis (14), Cell division and Cell cycle (15), DNA synthesis (16), Nucleotide metabolism (17), Cell organization (18), Cell wall synthesis/modification (19), Transport (20), RNA processing (21), Redox balance (22), Phyto-hormone metabolism (23), Protein modification (24), Lipids metabolism (25), Development (26), Secondary metabolism (27), Amino acid metabolism (28), Protein targeting and vesicle transport (29), Signaling regulation (30), Abiotic/biotic stress (31), Protein degradation (32), Regulation of transcription (33), Protein synthesis (34), Functional enzyme (35), and Unclassified (36). Proteins included in functional classification are provided in Appendix A.

The number of proteins clustered into these functional groups increased consistently for the six proteomes (Figure 3). Groups 1–16 each contains 8–100 proteins, group 17–27 each contains 101–200 proteins, group 27–29 each contains 201–300 proteins, Group 30–31 contains 201–300 proteins, Group 32–34 contains 301–400 proteins, Group 35 contains up to 500 proteins, and group 36 contains more than 1000 proteins. Ten functional groups (6-DNA repair, 7-Pentose phosphate pathway, 12-Mitochondrial electron transport/ATP synthesis, 13-TCA cycle, 16-DNA synthesis, 22-Redox balance, 25-Lipids metabolism, 27-Secondary metabolism, 28-Amino acid metabolism, and 30-Signaling regulation) each contains nearly identical number of proteins from the six proteomes. The following eight groups (3-Ammonia metabolism, 4-Fermentation, 5-Metal binding, 8-Carbohydrates metabolism, 9-RNA synthesis, 21-RNA processing, 24-Protein modification, and 33-Regulation of transcription) showed some disparity among the six proteomes. Based on the functional classification, the six proteomes all have a wide coverage of biological functions.

Then, we conducted the functional classification for the DAPs where each function group should contain a minimum of three proteins in one of the six proteomes. The DAPs were clustered into the following 26 functional groups: Abiotic/biotic stress (1), Functional enzyme (2), Development (3), Secondary metabolism (4), Protein degradation (5), Unclassified (6), Amino acid metabolism (7), Phyto-hormone metabolism (8), Signaling regulation (9), Cell organization (10), Nucleotide metabolism (11), Cell wall synthesis/modification (12), DNA synthesis (13), Lipids metabolism (14), Major CHO metabolism (15), Minor CHO metabolism (16), Protein targeting and vesicle transport (17), Redox balance (18), Regulation of transcription (19), RNA processing (20), TCA cycle (21), Transport (22), Glycolysis (23), Mitochondrial electron transport/ATP synthesis (24), Protein modification (25), and Protein synthesis (26). Proteins included in functional classification are provided in Appendix A.

First, the drought-up-regulated proteins in RT1-SDT was plotted in a decreasing order of protein number against the functional groups, and the remaining five proteomes were plotted following the sequence of functional groups (Figure 4). For the drought up-regulated proteins, the abiotic/biotic stress (group 1) represents the largest functional group (with the exception of the unclassified group for some proteomes) in all the six proteomes. For the remaining 24 functional groups (excluding the unclassified group), the number of proteins (for both drought-up- and down-regulated proteins) varied greatly across the six proteomes. A larger number of drought-up-regulated proteins were clustered into functional groups in all the proteomes with the exception of RT2-D1W where DAPs formed more drought-down-regulated functional groups. Functional classification of DAPs from the six proteomes indicate that these different types of root tissues expressed significant differences in individual proteins induced by drought and recovering treatments.

### 3.3. Drought-Induced Differentially Accumulated Proteins Involved in Phytohormone Perception and Signaling Network

The drought-induced DAPs involved in ABA signaling pathway include the pyrabactin resistance 1 (PYR1)/PYR1-like (PYL)/regulatory components of the ABA receptor (RCAR) (PYR/PYL/ RCAR) signalosome for ABA responses. In RT1 proteomes, three PYR1-like six proteins were found to be up-regulated (2.06–2.23-fold), whereas the ABI3 TF (0.46-fold) and an ABI3-interacting protein 3 were down-regulated (0.51-fold) in D3W. Two proteins including PYL8, and an ABA-responsive element binding protein 3 (AREB3) were up-regulated in D1W. The nodulin-related protein 1, which is annotated to a negative regulator of ABA signaling and synthesis, was up-regulated in D3W. In RT2 proteomes, the PYR1-like (0.27–0.40-fold, *p* < 0.05), a Malectin/receptor-like protein kinase (a negative regulator of the ABA signaling pathway), and an ABA-inducible HVA22 protein were down-regulated in SDT. The up-regulated DAPs include an ABA receptor protein (Pavir.Ba03030.1, 2.69-fold, *p* < 0.05), a cullin 3 with a possible role in degradation of ABA-inducible TFs (2.61-fold, *p* < 0.05), and the ABA-INSENSTIVE1 (ABI1) and ABI2 which are PP2Cs proteins (3.25–3.98-fold) acting as negative regulators of ABA signaling. Proteins involved in biosynthesis of ethylene, jasmoic acid (JA), auxin, gibberellic acid (GA), and the signaling pathways coordinated by these phytohormones were also identified (Table 2 and Table 3).

### 3.4. The Drought-Induced Differentially Accumulated Proteins (DAPs) Involved in Cellular Activities and Organogenesis

In RT1 proteomes, DAPs affecting cell division were down-regulated, which include microtubule-associated proteins, tetratricopeptide repeat (TPR)-like superfamily protein (anphase-promoting complex (APC) subunits cdc16, cdc23, and cdc27), and Centrin2 (required for centriole duplication and correct spindle formation). The NAP1-related protein 2, which acts as histone H2A/H2B chaperone in nucleosome assembly with a critical role for the correct expression of genes involved in root proliferation and patterning was down-regulated in SDT (0.23-fold), in D1W and D3W (0.43–0.48-fold). The annexin 5 involved in apoptosis was up-regulated in SDT (3.11-fold) and D1W (2.7-fold). A dormancy-associated protein (negative regulator of cell division mediated by the ethylene-induced TF) was up-regulated in SDT (2.82–2.88-fold) and D1W (4.43-fold). Two different aquaporin proteins associated with root cell proliferation were induced in SDT (2.95-fold) and D1W (2.43-fold). In D3W treatment, no significant differences in these proteins were identified between the drought-treated and the well-watered groups, which concurs with physiological properties of these plants in restoring to similar levels in leaf (Table 1). Several cell wall proteins were identified. In this group, the carrot EP3-3 chitinase homolog (Pavir.J17463.1, 10-fold) showed the highest level of increase in SDT, followed by a Dirigent-like protein (4.76-fold in SDT). A transducin family protein (annotated to LATERAL ROOT STIMULATOR 1) was down-regulated (0.37-fold) under drought treated condition.

In RT2 proteomes, seven proteins affecting lateral root meristem initiation (transducin/WD40 repeat-like superfamily protein) were up-regulated in SDT (2.54–3.13-fold). Three proteins for cell elongation growth (dynamin-like protein, varicose-related) (3.65–8.67-fold) were up-regulated and two aquaporin proteins were down-regulated (0.38-fold) in the SDT, D1W, and D3W. Proteins for cell wall loosening such as expansin, and biosynthesis of cell wall component materials (cellulose, xylan, pectin, and lignin) were identified, and a majority of these proteins were up-regulated. A tonoplast-localized protein required for secondary wall formation, Walls Are Thin 1, was up-regulated by 10-fold. Proteins for secondary wall formation were among the up-regulated proteins which include cellulose synthase (4.22-fold), alpha-xylosidase (3.46-fold), and fasciclin-like arabinogalactan family protein (6.39-fold) in the SDT treatment. In D3W three expansin proteins were up-regulated (Pavir.Ib03630.1, Pavir.Ib02222.1, Pavir.Bb01836.1; 2.18–2.39-fold), which concurs with the cell activity of root lengthening growth upon re-watering.

### 3.5. The Drought-Induced Differentially Accumulated Proteins (DAPs) Involved in Tolerance/Resistance to Physiological Stresses in Roots

From RT1 and RT2 proteomes, a large number of hydrophilic proteins including dehydrins, late embryogenesis abundant proteins (LEA), osmotins, and several regulatory proteins for drought stress, were up-regulated. These proteins play positive roles in plant defense against water DAPrivation within cells. In general, the SDT stage was identified with the largest number of proteins in this group, which also showed the greatest magnitude of increases in the abundance level. Several proteins maintained as up-regulated in D1W and D3W, and new proteins (different protein accessions but conferring the same function) also appeared during the recovery stage. Antioxidant enzymes include ascorbate oxidase, peroxidases, plant thionin, and thioredoxin, were all up-regulated under the drought-treated condition. A D-arabinono-1,4-lactone oxidase involved in ascorbate synthesis was up-regulated but with decreasing magnitude from water-stressed through the recovery stages (7.69- in SDT, 6.63-fold in D1W and 3.02-fold in D3W). The detoxification mechanism involves DAPs of glutathione peroxidase, glutathione S-transferase, glutathione reductase for the maintenance of the glutathione antioxidant pool, the glyoxalase for the degradation of toxic methylglyoxal, the metallothionein for detoxification of metal ions and reducing oxidative stress. These proteins were all up-regulated under drought-treated conditions.

### 3.6. The Drought-Induced Differentially Accumulated Proteins (DAPs) Involved in Gene Transcription and Protein Translation Activities

In RT1 proteomes under the SDT treated condition, ribonucleases involved in post-transcriptional regulation (2.34–7.66-fold), a homeodomain-like superfamily protein (3.05-fold), and NmrA-like negative transcriptional regulator family protein (2.92-fold) were up-regulated. The down-regulated DAPs include several TFs including bHLH (0.17–0.37-fold), PHD (0.37-fold), RAD-like 1 (0.25-fold), squamosal promoter binding protein-like 9 (0.31-fold), and winged-helix DNA-binding transcription factor family protein (0.29-fold). A transcription activator was up-regulated in SDT (9.73-fold) and D1W (8.95-fold). Seven TFs were all down-regulated in D1W (0.24–0.45-fold), but the multi-protein bridging factor 1A (transcriptional coactivator) was up-regulated under the same treatment condition. In D3W, four TFs were down-regulated. Two transcription elongation factors (TFIIS, TFIIE) were down-regulated in D1W and D3W.

In RT2 proteomes, the up-regulated DAPs in SDT affecting gene transcription include TCP transcription factor (TF) (8.12-fold), the TBP-associated factors (TAFs) (3.10-fold), and a transcription regulator (3.10-fold), and the down-regulated DAP is the nuclear factor Y, subunit B11 TF (0.24-fold). The multi-protein bridging factor 1A (transcriptional coactivator) was up-regulated in D1W. The general regulatory factor 7 proteins (the TF for fatty acid synthase genes) were down-regulated in D1W in RT2 and D3W in RT1 (0.29–0.50-fold).

For protein translation, in RT1, the down-regulated DAPs include two eukaryotic translation initiation factors (0.35–0.38-fold) in SDT; the nuclear transport factor 2 (NTF2) family protein with RNA binding (RRM-RBD-RNP motifs) domain in D1W and D3W (0.41–0.42-fold), and 14 proteins (0.400.50-fold) including eukaryotic release factor 1–3, eukaryotic translation initiation factors, translation elongation factor EF1B, translation initiation factor IF2/IF5, GTP binding elongation factor Tu family protein, and essential protein Yae1. DAPs identified in RT2 include a eukaryotic translation initiation factor (2.78-fold), two eIF4AIII binding proteins with opposite changes (0.26-fold, 8.26-fold), translation machinery associated TMA7 (0.30-fold), and the translation protein SH3-like family protein (6.58-fold).

A number of ribosome proteins were down-regulated in RT1 and RT2 in SDT. During recovery stages, these down-regulated DAPs remained low abundance in D1W, recovered in D3W in RT1, none of these proteins showed significant changes in RT2. These results indicate that the protein translation activity was more affected in root tips during the water-stressed and the first day of water recovery. By the D3W, as tissues recovered from the stress conditions, the protein translation activity was restored. More importantly, protein translation was restored more promptly in RT2 than RT1 upon re-watering.

### 3.7. The Drought-Induced Differentially Accumulated Proteins (DAPs) Involved in Metabolic Pathways

Under water deficit conditions, a decrease in carbon assimilation is the most common physiological phenomenon in plants. The partition of a reduced amount of sucrose transported into roots is determined by two sucrolytic enzymes, sucrose invertase (INV; EC 3.2.1.26), and sucrose synthase (SuSy; EC 2.4.1.13). Invertase catalyzes the irreversible reaction to convert sucrose into glucose and fructose. Sucrose synthase catalyzes the reversible reaction that converts sucrose into fructose and either uridine diphosphate glucose (UDP-G) or adenosine diphosphate glucose (ADP-G). In RT1, the sucrose invertase was down-regulated in SDT (0.34-fold), but two sucrose synthases were up-regulated in SDT (3.37-fold), D1W (2.51–3.66-fold) and D3W (2.42–2.58-fold). Two proteins in the pentose shunt pathway were up-regulated, the transketolase (2.78-fold) in SDT and aldolase (2.23-fold) in D1W. Six DAPs in glycolysis were up-regulated as follows: fructose-bisphosphate aldolase 2 and phosphoglycerate kinase in SDT (2.71–2.86-fold), phosphoglycerate kinase and phosphofructokinase 2 in D3W (2–2.57-fold), and phosphoglycerate kinase, phosphofructokinase 2, phosphoglycerate kinase, phosphoglycerate mutase, 2,3-bisphosphoglycerate-inDAPendent in D1W (2.31–5.40-fold). The aconitase 3 in TCA cycle was up-regulated in SDT (4.72-fold); the pyruvate metabolic pathway enzymes (pyruvate orthophosphate dikinase, phosphoenolpyruvate carboxylases) were repressed in SDT and D1W. The Melibiase which catalyzes the hydrolysis of melibiose into galactose and glucose were up-regulated in SDT (6.02-fold), D1W (4.41-fold), and D3W (2.93-Fold).

In RT2, a neutral invertase was down-regulated (0.25-fold). The sugar transporter 1 (H + / H + /monosaccharide cotransporter (STP1)) which is involved in sugar signaling was up-regulated (nine-fold). In the same tissue, one pentose shunt enzyme (aldolase) was up-regulated (2.8-fold) in SDT and D3W. Several proteins in glycolysis pathway were up-regulated in SDT and D1W, and those in TCA were all down-regulated which include pyruvate dehydrogenase E1 alpha (0.18–0.35-fold) and fumarase 1 (0.39-fold) in SDT, and pyruvate orthophosphate dikinase, phosphoenolpyruvate carboxylase in D1W and D3W (<0.50-fold).

Drought also induced changes in proteins in amino acid metabolic pathways. In RT1, the glutamate decarboxylase in GABA shunt, and homogentisate 1,2-dioxygenase involved in the catabolism pathway of tyrosine were induced in SDT (3.69–4.80-fold) and D1W (3.69- and 2.84-fold). In the D1W, the dihydrodipicolinate synthase 1 for lysine biosynthesis was upregulated (2.62-fold), and the pyrroline-5-carboxylate (P5C) reductase for arginine and proline metabolism was repressed (0.51-fold). In RT2 under SDT treatment, the up-regulated DAPs include several rate limiting enzymes for amino acids biosynthesis. These proteins include the imidazoleglycerol-phosphate dehydratase for histidine biosynthesis, the Class-II DAHP synthetase family protein for aromatic amino acids biosynthesis, the glutaminyl cyclases (QCs) catalyzing the formation of pyroglutamate (pGlu) to store glutamate. The down-regulated DAPs include S-methyl-5-thioribose kinase for methionine synthesis (0.29-fold). For the branched side chain amino acids (BCAAs) metabolism, the ketol-acid reductoisomerase for the biosynthesis of BCAAs was down-regulated (0.38-fold) whereas the branched-chain amino acid transaminase 5 for degradation of BCAA was up-regulated (4.68-fold). The action of these two enzymes could lead to lower BCAA content in RT2. In D1W, glutamate decarboxylase for GABA shunt for the decarboxylation of glutamic acid as a means to avoid reactive oxygen species (*ROS*) accumulation (3.71-fold) and glutamine synthase (2.29-fold) were both up-regulated, which indicate an active catabolism of glutamate in this tissue. In D3W, the chorismate mutase for aromatic AA biosynthesis was up-regulated (2.06-fold), and the down-regulated DAPs include the D-3-phosphoglycerate dehydrogenase for L-serine, and O-acetylserine (thiol) lyase (also named cysteine synthase) for cysteine (0.45–0.48-fold). Additional DAPs involved in metabolic pathways of carbohydrates, amino acids, and fatty acids as well as secondary metabolites are provided in Table 2 and Table 3.

### 3.8. Access to Mass Spectrometry Proteomics Data

The mass spectrometry proteomics data have been DAPosited to the ProteomeXchange Consortium via the PRIDE partner repository with the dataset identifier PXD017441 and 10.6019/PXD017441. The project name is: “Comparative Proteomics of Root Apex and Root Elongation Zones Provides Insights into Molecular Mechanisms for Drought Stress and Recovery Adjustment in Switchgrass” (https://www.ebi.ac.uk/pride/).

## 4. Discussion

The plant root apical region extends from the root apex to the maturation zone where root hair initiates, it is comprised of layers of cells of different identities in transverse as well as vertical directions. Each of these types of cells performs distinct functions in modifying root architectural system and withstanding the suboptimal environmental conditions. As cells in these different root zones carry out distinct physiological and cellular functions, they should express different proteomes under normal as well as suboptimal conditions. In this study, we have conducted proteomics analysis of two root sections each representing the root apical root cap/root meristem/differentiation, and the cell expansion/cell maturation zones during water-stressed conditions and recovery process. Proteomics analysis results showed that the two root sections differed significantly in terms of composition of proteomes and the drought-induced DAPs. Under water stressed conditions (SDT), only a few DAPs (nine proteins out of 674 DAPs) overlapped between the two root sections. Upon re-watering, the percentage of overlapping DAPs increased up to 27.6% (155/562) after one day of re-watering (D1W), however, it was decreased to 15% (84/565 total) in D3W. Furthermore, a majority of these DAPs were identified only in one of the three treatments. Such proteomic changes agree with previous studies indicating a very small number of proteins and genes overlapped spatially in different root regions under drought stress, and temporally from drought treatment to recovery adjustments phases [40,41].

However, when these DAPs were compared in the context of biological functional categories, more commonalities were identified between the two root sections. Among the eight protein functional groups showing the same trend of changes in the two root sections, the hydrophilic proteins (LEA, dehydrin, osmotins, etc.) and antioxidant enzymes were all up-regulated under drought condition. Cellular enrichment of these hydrophilic proteins under water deficit conditions play a key role in maintaining cellular water status, and thus help to prevent the desiccation-induced aggregation of the water-soluble proteomes [42]. A recent study, on the leaf proteomes of drought treated wheat (*Triticum aestivum* L.), showed that the abundances of over half of the highly hydrophilic LEA proteins changed significantly after drought stress and they were involved in protection against drought [43]. One of the primary changes in response to drought is the generation of reactive oxygen species (ROS). The ROS level is controlled in two ways: the first by the removal under the action of the antioxidant enzymes like SOD, catalase, and peroxidases and the antioxidant compounds like ascorbate and reduced glutathione, and the second by changing metabolic pathways to control the generation of ROS. In this study, several antioxidant enzymes were induced in the two root sections, and the pentose phosphate pathway was also induced which provide a major source of NADPH reducing equivalents, repressing the TCA pathway as a means of reducing oxidative reactions [44,45]. Together, these results indicate that induction of hydrophilic proteins and antioxidant enzymes is a universal response to drought stress across all plant cells types.

A large number of proteins involved in cell wall remodeling were identified in the two root sections. These cell wall DAPs include expansins, laccase, and hydroxyproline-rich glycoprotein. Expansins are important proteins for cell wall loosening [46]. According to Hanetal [47], overexpressing a wheat β-expansin (*TaEXPB23*) in tobacco enabled the plants to have a higher water retention ability and develop a more vigorous root system under high salt conditions than wild type seedlings [48]. Laccases are copper-containing oxidase enzymes. They play a role in the formation of lignification of the secondary cell walls by promoting the oxidative coupling of monolignols [49]. Hydroxyproline-rich glycoproteins (HRGs) is a major group of wall glycoproteins, and the mobile soluble HRGs are considered to play a key role in modulating cell-cell communication [50]. A study on rice showed that the HRG proteins were induced by ABA and stress, and in turn by interacting with cell wall components near the plasma membrane to act as suppressor of root cell expansion [51]. Results from this study showing the enrichment of these cell wall SCPs under-water stressed condition provide more evidences for the importance of these cell wall proteins in drought tolerance.

Under stress conditions, the repression of global protein synthesis (reduction of eIF4F and eIF2, ribosomal proteins), and selective translation of mRNAs encoding proteins that are vital for cell survival and stress recovery are important for tolerance [52]. In the two root sections, DAPs of the protein translation machinery members were down-regulated under drought-treated conditions which concurred with the previous study.

According to the DAPs, the two root sections also exhibited differential responses to water-stressed conditions. These reactions are more related to the physiological functions of the respective root regions. In the root tip section, the drought-repressed DAPs in mitosis include the TPX2 (targeting protein for Xklp2) protein which is essential for nuclear envelope breakdown and initiation of prospindle assembly [53,54], the MAD2 protein (Pavir.J13426.1) with a key role early mitosis. Root tip cell elongation mediated by the cell wall loosening proteins, is essential for developing extended root system thus conferring drought tolerance [46,55]. Several proteins affecting cell wall loosening (expansin) increased significantly across the two root sections and consistently throughout the water-stressed and at the 1- and 3-day recovery stages. Taken together, the switchgrass roots may have favored a mechanism of reducing mitotic cell division, while allocating resources to cell elongation under water-stressed condition, and the cell elongation proteins are mobilized to enable fast root growth when adequate water sources become available. Such mechanism might be related to the octoploid polypoidy genome of Switchgrass polyploids [56] as mitosis of the polyploidy cells are prone to errors under stress conditions. The switchgrass, by reducing mitosis, can avoid expenses of energy in this error-prone process, instead, the plants can allocate more of the limited resources to increase cell sizes, to maintain a root system as the drought tolerance mechanism [57].

A long-period of drought stress has wide-spread impact on plants from root to the above-ground tissues. In a previous proteomics study on drought-treated leaves of switchgrass, we have identified that the ABA signaling pathway was mobilized to defend against the stress. Proteomics changes involve in global transcription and translation activities, and abundance of enzymes affecting metabolic pathways of carbohydrates, amino acids, and fatty acids also expressed significant change under drought-treated conditions [23]. When integrated with root DAPs identified in this study, it is clear that the ABA signaling pathway was activated by drought stress but involving different proteins in leaf and roots. Similarly, proteins within the same biological process or functional pathway were found to be affected differently in roots and leaves. The identification of these individual proteins will aid in the selection of important proteins (genes) for future research to develop drought tolerant plants.

## 5. Conclusions

The proteomics study has identified a list of drought-induced DAPs in two root sections (the basal 1-cm root-tip and the upper 1-cm elongation and maturation zones), under water-stressed treatment and the recovery phases after re-watering for one and three days. The drought-induced DAPs were classified into three groups based on trend of drought-induced change. The first group of proteins expressed changes following the same trend across the two root sections. These DAPs include hydrophilic proteins such as LEA, dehydrin, osmotin, and proteins involved in cell wall remodeling and cellular detoxification mechanisms, and antioxidant enzymes. These proteins provide drought tolerance/resistance mechanisms at the whole root level. The second group of DAPs showed opposite changes between the two root sections, such as those involved in ethylene and ABA metabolism and signaling. The fine tuning of the relative balance of these stress hormones may have a role in activating stress-genes as well as organogenesis processes in different root zones. The third group of DAPs includes proteins unique to each individual root section and treatment conditions. These proteins are more relevant to the induction and suppression of root secondary organogenesis (root hair, lateral roots) and root tip orientation. Additionally, different proteins isoforms were induced according to the drought treatment conditions, and the respective individual DAP (and the encoding genes) should be carefully evaluated for functional studies of plant tolerance to drought. In summary, this study has clearly shown a complex and fine-tuned proteomics regulation for root development under water-stressed conditions and prompt recovery to well-watered conditions in Switchgrass. The identified proteome changes concur with high drought tolerance of the plants.

## Figures and Tables

**Figure 1 proteomes-08-00003-f001:**
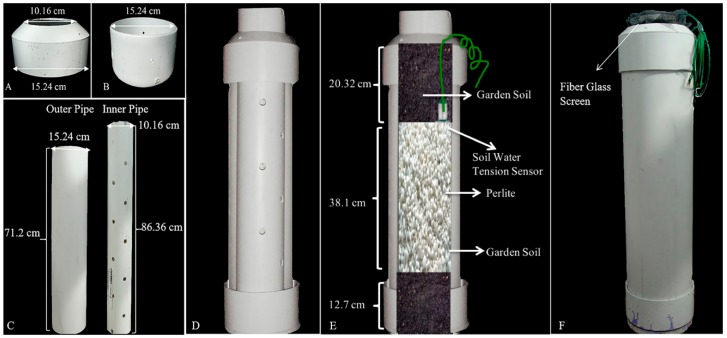
Structure of the “sandwich-type” drought treatment system. (**A**) PVC sewer and drain coupling; (**B**) PVC sewer and drain cap; (**C**) Double PVC pipes: outer pipe and perforated inner pipe; (**D**,**E**) Perspective of “sandwich” drought system; (**F**) Combination of “sandwich” drought system structure.

**Figure 2 proteomes-08-00003-f002:**
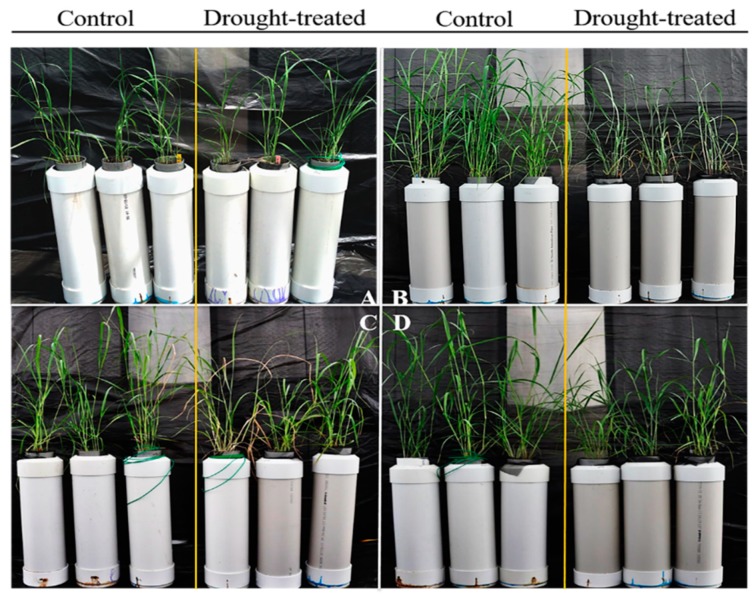
Phenotypic changes of switchgrass under different levels of drought treatment. (**A**) Plants under normal watering condition (4 L of water every 3-days) before drought treatment. (**B**) The drought-treated plants with gradual drying process and control plants with normal watering at the 29th day, respectively. (**C**) The drought-treated plants with 1-day re-watering after 29 days gradual drying process and control plants with continuous normal watering, respectively. (**D**) The drought-treated plants with 3-day re-watering after 29 days gradual drying process and the control plants were watered regularly.

**Figure 3 proteomes-08-00003-f003:**
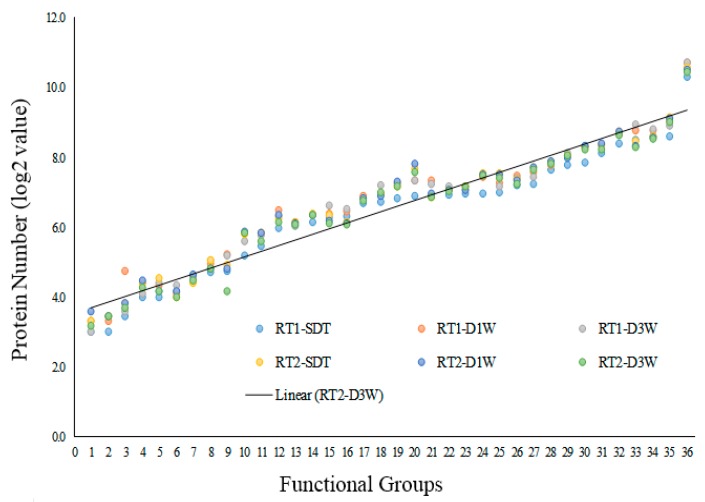
Functional categorization of proteins identified in root-tips of switchgrass plants under severe drought treatment (SDT), 1-day re-watering (D1W) and 3-day re-watering (D3W). RT1: basal 1-cm root-tip (RT1) tissues; RT2: elongation/maturation region of root-tips. Functional pathways were constructed using Mapman.

**Figure 4 proteomes-08-00003-f004:**
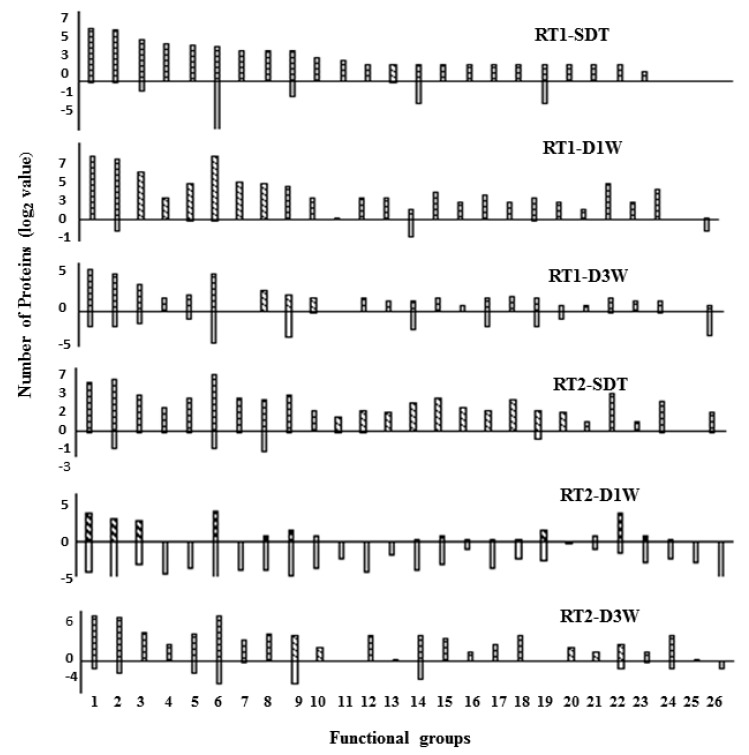
Functional categorization of drought-induced differentially accumulated proteins (DAPs) identified in root-tips of switchgrass plants under severe drought treatment (SDT), 1-day re-watering (D1W) and 3-day re-watering (D3W). RT1: basal 1-cm root-tip (RT1) tissues; RT2: elongation/maturation region of root-tips. Functional pathways were constructed using Mapman. Only groups containing three and more proteins in one of the six treatments are presented here. Negative value indicates the number of proteins in each group belonging to stress-down-regulated proteins, and positive values are for to stress-up-regulated proteins.

**Table 1 proteomes-08-00003-t001:** Effects of drought treatments on physiological properties of switchgrass plants.

Levels of Drought Condition	Treatment	Leaf Photosynthetic Rate (μmol CO_2_ /m^2^/s)	Stomatal Conductance (mol H_2_O /m^2^/s)	Transpiration (mmol H_2_O/m^2^/s)	Water Use Efficiency (μmol CO_2_ /mmol H_2_O)
Level 1 (Before drought treatment)	Control	23.31 ± 4.59 ^A^	0.14 ± 0.03 ^A^	4.44 ± 1.13 ^A^	5.40 ± 0.99 ^A^
	Drought	22.54 ± 3.97 ^A^	0.14 ± 0.03 ^A^	4.25 ± 1.03 ^A^	5.46 ± 0.99 ^A^
Level 2 (28 days drought)	Control	22.02 ± 4.06 ^A^	0.15 ± 0.03 ^A^	5.38 ± 1.23 ^A^	4.16 ± 0.56 ^A^
	Drought	16.62 ± 5.82 ^B^	0.12 ± 0.05 ^B^	4.51 ± 1.54 ^B^	3.77 ± 0.75 ^B^
Level 3 (28 days drought + 1 day re-watering)	Control	18.98 ± 4.93 ^A^	0.12 ± 0.03 ^A^	3.64 ± 0.87 ^A^	5.21 ± 0.64 ^A^
	Drought	15.10 ± 4.99 ^B^	0.10 ± 0.03 ^B^	2.67 ± 0.93 ^B^	5.83 ± 1.33 ^B^
Level 4 (28 days drought + 3 days re-watering)	Control	12.48 ± 2.91 ^A^	0.18 ± 0.06 ^A^	2.21 ± 0.64 ^A^	5.80 ± 0.82 ^A^
	Drought	11.54 ± 4.2 ^A^	0.15 ± 0.05 ^B^	2.0 ± 0.67 ^A^	5.80 ± 0.86 ^A^

Data are represented by the means of all the replicate plants. Within the same level of drought treatments, the means in the same column followed with the same number (A, or B) are not significantly different between drought-treated and control groups (*p* < 0.01) when tested using ANOVA.

**Table 2 proteomes-08-00003-t002:** Classification of biological processes for drought-induced differential accumulated proteins in root-tip elongation/maturation zone (RT2) tissues ^a^

Biological Process	Protein Accessions ^b^	Description	Fold Change (Drought Treated/Control) ^c^		
			SDT ^d^	D1W ^e^	D3W ^f^
Phytohormones ABA	Pavir.Ab00822.1 Pavir.Da00781.1	PYR1-like 2 (ABA RECEPTOR1)	0.27–0.40		
	Pavir.Ab01425.1	Malectin/receptor-like protein kinase family protein (negative regulation of abscisic acid-activated signaling pathway)	0.33		
	Pavir.Ba03030.1	RNA binding;abscisic acid binding (ABA receptor)	2.69		
	Pavir.Db00930.1	ABA- and stress-inducible osmotic stress tolerance gene (HVA22 homologue)	0.26		
	Pavir.Aa00702.1	Cullin 3	2.61		
	Pavir.Gb00126.1	IQ-domain 32		0.39	
	Pavir.Ab01039.1	REGULATORY COMPONENTS OF ABA RECEPTOR3 (RCAR3, PYL8)		3.58	
	Pavir.J34085.1	ABA-RESPONSIVE ELEMENT BINDING PROTEIN 3, AREB3		2.51	
	Pavir.Ia02723.1	Nodulin-related protein 1			2.17
	Pavir.Ba01419.1; Pavir.Gb00423.1	Protein phosphatase 2C family protein	3.25–3.98		
Auxin	Pavir.Fa00409.1	Auxin-responsive family protein	0.33		
	Pavir.Cb00986.1	IBA response protein	3.54		
	Pavir.Fb00898.1	Auxin-responsive family protein	6.58		
	Pavir.Ia02169.1	Flavin-binding monooxygenase family protein (Auxin biosynthesis)	0.40		
Ethylene	Pavir.Fb01856.2; Pavir.Ba01308.1;	S-adenosyl-L-methionine-dependent methyltransferase	2.86–3.78		
	Pavir.Fa00353.1			0.35	
	Pavir.J10341.1;	S-adenosylmethionine synthetase	3.31		
	Pavir.J19251.1			0.36	
	Pavir.Ea01573.1	ACC oxidase 1	2.88	0.29	
	Pavir.J10323.1			0.31	0.45
	Pavir.Ib00921.1	Ethylene response regulator 2	0.21		
	Pavir.Ib03775.1	Dormancy-associated protein (Ethylene inducible TF (ERF114) down-regulated cell proliferation		2.80	
	Pavir.J19751.1			3.86	
JA	Pavir.Ca00561.1	12-oxophytodienoate reductase 1	0.31		
	Pavir.Da01421.1	12-oxophytodienoate reductase 1	0.37		
Lateral root initiation	Pavir.Bb03551.1	Transducin/WD40 repeat-like superfamily protein	2.54		
	Pavir.Bb01815.1		2.66		
	Pavir.Ba00100.1		2.98		
	Pavir.J08344.1		3.39		
	Pavir.Cb02019.1		3.41		
	Pavir.Fa01415.1		4.04		
	Pavir.J02298.1		3.13		
Aquaporin associated root meristem cell division	Pavir.Gb01084.1	plasma membrane intrinsic protein	0.39		
	Pavir.Bb01320.2		0.38		
	Pavir.J11885.1			0.38	0.47
Cell elongation growth	Pavir.Ca00886.1	dynamin-like protein	3.65		
	Pavir.Da01210.1	dynamin-like protein 6	4.43		
	Pavir.Ab03037.1	varicose-related	8.67		
Cell wall	67 proteins	Expansin; Pectin, Xylan, Cellulose, Fasciclin, Chitinase, Arabinogalactan, Lignin, hydroxyproline-rich glycoprotein, Other cell wall components	2.86–9.68 (14 proteins) 0.33–0.39 (3 proteins)	2.21–4.71 (19 proteins); 0.20–0.39 (6 proteins)	2.01–5.01(22 proteins); 0.40–0.46 (2 proteins)
Water stress proteins (Dehydrins, LEA, Osmotic)	Pavir.J35962.1	Dehydrin	0.39		4.11
	Pavir.Gb00029.1	Early-responsive to dehydration stress protein (ERD4)	5.31		
	Pavir.Gb00029.1	Responsive to dehydration 21	2.69		
	16 proteins	LEA		3.54–11.07	2.11–9.25
	6 proteins	Osmotin		2.2–3.18	4.02–6.66
	Pavir.J37770.1	spermidine synthase 1			
Oxidative stress	Pavir.J22922.1	monodehydroascorbate reductase 1	0.38	2.19	
	16	peroxidase	2.62–17.67 (5)	0.26–7.17 (11)	2.07–3.55 (6)
	Three proteins	Plant L-ascorbate oxidase	13.73	2.15	2.33–2.41
	Pavir.Ib02577.1 Pavir.Hb01571.1	Plant thionin		2.29–3.87	2.79–4.62
	Pavir.Bb00430.1; Pavir.J05150.1; Pavir.J05150.1	thioredoxin H-type 1	2.61	2.28	2.06
Detoxification	26 proteins	glutathione peroxidase 1; glutathione S-transferase; glyoxalase; metallothionein; glutathione reductase	2.71–3.98	2.28–6.10	2.01–4.03
Transcriptional Regulation (TFs)	Pavir.J05134.1	TCP family transcription factor (TF)	8.12		
	Pavir.J01608.1	TBP-associated factor II 15 (TF)	3.10		
	Pavir.Fa01472.1; Pavir.Fb01387.1	multiprotein bridging factor 1A (Transcriptional coactivator)		3.82–3.61	
	Pavir.Fa00854.1; Pavir.J16866.1; Pavir.Aa01534.1	general regulatory factor 7 (TF for fatty acid synthase genes)		0.35–0.36	0.43–0.45
	Pavir.Ib02226.1	transcription regulators	3.90		
	Pavir.Bb01701.1	nuclear factor Y, subunit B11	0.24		
	Pavir.Ia04404.1	transcription activator			6.31
	Pavir.Ha01744.1	Transcription elongation factor (TFIIS) family protein			0.50
Protein translation	Pavir.Ba00102.1	Ribosomal protein L10	3.03		
	Pavir.Cb01702.1	Ribosomal protein L13	2.91		
	Pavir.Ea00990.1	Ribosomal L29e	4.63		
	Pavir.Ea02098.1	Ribosomal protein L25	2.51		
	Pavir.Ea02335.1	ribosomal protein L12-A	5.12		
	Pavir.Fa00399.1	Ribosomal protein L32e	2.65		
	Pavir.Fb00702.1	Ribosomal protein L7Ae	2.94		
	Pavir.Gb02168.1	Ribosomal protein L10	4.07		
	Pavir.Ib01816.1	Ribosomal protein S4	0.38		
	Pavir.J00318.1	ribosomal protein L24	0.38		
	Pavir.J04126.1	Ribosomal protein S3	6.41		
	Pavir.J14971.1	Ribosomal protein S21e	2.52		
	Pavir.J21269.1	Ribosomal protein L6 family	4.85		
	Pavir.J33201.1	Ribosomal protein S4 (RPS4A)	0.40		
	Pavir.J39391.1	Ribosomal protein L31e	4.85		
	Pavir.Eb01973.1	CASC3/Barentsz eIF4AIII binding	0.26		
	Pavir.Ca01872.1		8.26		
	Pavir.Bb02918.1	Eukaryotic translation initiation factor	2.78		
	Pavir.Da00112.1	Translation machinery associated TMA7	0.30		
	Pavir.Eb02299.1	Translation protein SH3-like family protein	6.58		
	Pavir.J11476.1 Pavir.J16816.1	Eukaryotic translation initiation factor 3 subunit 7 (eIF-3)		0.48–0.49	
	Pavir.J01318.1	Eukaryotic translation initiation factor 4A1		0.39	
Amino acids	Pavir.J17001.1	S-methyl-5-thioribose kinase (Methionine synthesis)	0.29		
	Pavir.Ca00973.1	Ketol-acid reductoisomerase (valine, leucine and isoleucine)	0.38		
	Pavir.J20971.1	Imidazoleglycerol-phosphate dehydratase (histidine)	2.76		
	Pavir.Ib01852.1	Class-II DAHP synthetase family protein (aromatic amino acids)	2.88		
	Pavir.Da02432.1	Glutaminyl cyclase	3.02		
	Pavir.Fb01711.1	Glutamate decarboxylase (GABA shunt)		3.71	
	Pavir.Ib02399.1	Branched-chain amino acid transaminase 5 (BCAT5)	4.68		
	Pavir.Ia00863.1	Glutamine synthase clone F11		2.29	
	Pavir.J14758.1	D-3-phosphoglycerate dehydrogenase (L-serine)			0.45
	Pavir.J13435.1	O-acetylserine (thiol) lyase (OAS-TL) isoform A1 (cysteine)			0.48
	Pavir.Ab00560.1	Chorismate mutase 2 (aromatic AA)			2.06
Carbohydrate metabolism	Pavir.J20347.1	Neutral invertase	0.25		
	Pavir.Ia03415.1	Sucrose synthase 3		2.29	
	Pavir.Ib01813.1	Sucrose synthase 4		0.36	
	Pavir.J19702.1	Sucrose synthase 6		2.60	
	Pavir.Bb00094.1	Sugar transporter 1	9.72		
	Pavir.Ba01978.1	Phosphoglucose isomerase	9.72		
	Pavir.J01199.1	Starch branching enzyme	3.88	2.76	
	Pavir.J06227.1; Pavir.Ca02850.1	Pyruvate dehydrogenase E1 alpha	0.18–0.35		
	Pavir.Ia03418.1	Fumarase 1	0.39		
	Pavir.J02207.1; Pavir.J15849.1	Pyruvate orthophosphate dikinase		0.38	0.23–0.29
	Pavir.Ga01969.1; Pavir.Gb01988.1; Pavir.J33969.1	Phosphoenolpyruvate carboxylase		0.38	0.45–0.48
	Pavir.Ib01540.1	Phosphoglycerate mutase			4.46
	Pavir.Fa02320.1; Pavir.J02390.1	glyceraldehyde-3-phosphate dehydrogenase C subunit 1	3.25–4.13		
	Pavir.Ga01407.1	Phosphofructokinase 2		3.08	
	Pavir.Ib01540.1	Phosphoglycerate mutase		6.11	
	Pavir.Db00351.1	Aldolase-type TIM barrel family protein	2.8	2.85	2.6
	Pavir.Ba00222.1	Melibiase family protein			2.72

^a^ Proteins with a fold change value (from drought-treated to non-treated control groups) greater than two standard deviations at adjusted *p* < 0.05 and identified with a minimum of two unique peptides. ^b^ Protein accession in switchgrass annotated database Pvirgatum_v1. 1. ^c^ Protein fold change from drought treated to non-drought treated groups. ^d^ Proteins from the severe drought-treated experiment. ^e^ Proteins from the 1-day re-watering experiment. ^f^ Proteins from the 3-day re-watering experiment.

**Table 3 proteomes-08-00003-t003:** Classification of biological processes for drought-induced differential accumulated proteins in basal root-tip (RT1) tissues ^a^

Biological Process	Protein Accessions ^b^	Description	Fold Change (Drought Treated/Control) ^c^		
			SDT ^d^	D1W ^e^	D3W ^f^
Phytohormones	Pavir.Ia03517.1	PYR1-like 6			2.17
	Pavir.Ib01400.1				2.23
	Pavir.J39695.1				2.06
	Pavir.Gb00126.1	IQ-domain 32			0.33
	Pavir.Ia01238.1	ABI3-interacting protein 3 (negative regulator of ABA signaling, PP2C)			0.51
	Pavir.Ca02694.1	ABI3 (AP2/B3-like) TF			0.46
	Pavir.Ia02723.1	Nodulin-related protein 1 (a negative regulator of the ABA signaling/synthesis)	3.04	3.14	1.70
	Pavir.Ca02736.1	IAA-induced protein 16	0.38		
	Pavir.J10323.1; Pavir.Ea01573.1	ACC oxidase 1	0.38	0.38–0.39	
	Pavir.J06494.1	Gibberellin-regulated family protein	4.15	4.81	
Root organogenesis (meristem cell division)	Pavir.Bb01893.1	Annexin 5 (apoptosis)	3.11	2.7	
	Pavir.Db00940.1 Pavir.Aa00752.1	microtubule-associated proteins	0.36–0.37		
	Pavir.Ib03538.1	Tetratricopeptide repeat (TPR)-like superfamily protein (anphase-promoting complex (APC) subunits cdc16, cdc23 and cdc27)	0.34		0.50
	Pavir.J21702.1		0.32		
	Pavir.Ib03761.1			0.44	
	Pavir.J29419.1	Centrin2 (Required for centriole duplication and correct spindle formation)			0.49
	Pavir.J28355.1	Methionine sulfoxide reductase (a central regulator of cell proliferation and differentiation)	0.35		
	Pavir.Aa01553.1	NAP1-related protein 2 (Acts as histone H2A/H2B chaperone in nucleosome assembly, playing a critical role for the correct expression of genes involved in root proliferation and patterning)	0.23	0.43	0.48
	Pavir.J19751.1	Dormancy-associated protein (ethylene induced-negative regulator of Cell division from TF ERF114);	2.82	4.43	
	Pavir.Ib03775.1		2.88		
	Pavir.Ga01149.1	Aquaporin for sustaining root meristem cell division)	2.95		
	Pavir.Gb00671.1			20.43	
Cell wall proteins	21 proteins in SDT; 24 proteins in D1W; 22 proteins in D3W	Beta-1,3-glucanase; Beta-hexosaminidase; chitinase; d-arabinono-1,4-lactone oxidase; Expansin; Laccase; Hydroxyproline-rich glycoprotein	2.68–10.34	20.40–6.80	1.98–3.91
Organogenesis and differentiation	Pavir.Ba01536.1	Root hair specific 19	0.37		
	Pavir.J03344.1	Root hair initiation protein root hairless 1 (RHL1)		0.49	0.49
	Pavir.Eb02336.1	Dirigent-like protein (Castrip)			
	Pavir.Ha00936.1		4.76		0.47
	Pavir.Aa03031.1; Pavir.J12222.1	TPX2 (wvd2) protein (The wvd2 gain-of-function mutant has impaired cell expansion and root waving, and changed root skewing)	0.36–0.38		
	Pavir.Ba01202.1	Transducin family protein / WD-40 repeat family protein (LATERAL ROOT STIMULATOR 1)	0.36		
	Pavir.Bb02689.1 Pavir.Ca00095.1 Pavir.Ha01893.1 Pavir.Ia01197.1				0.36–0.50
Water stress proteins (Dehydrins, LEA, Osmotins)	Pavir.Cb00662.1	Dehydrin	6.39	4.82	4.68
	Pavir.J13075.1			20.51	
	Pavir.Ea02542.1			5.86	
	Pavir.Aa00887.1				0.47
	Pavir.J04551.1				0.46
	Pavir.J13075.1				7.88
	Pavir.Aa02737.1	LEA	6.14	4.86	3.09
	Pavir.Bb02409.1		3.71	3.95	2.95
	Pavir.Ca01780.1		12.15	5.36	4.90
	Pavir.Eb02512.1		3.23	3.02	3.19
	Pavir.Ga00596.1		6.30	5.85	6.82
	Pavir.Gb00543.1		10.76	7.06	7.80
	Pavir.Ia03678.1		6.29	4.13	3.46
	Pavir.J00158.1		21.57	14.23	6.24
	Pavir.J28600.1		8.29	9.69	6.55
	Pavir.J24821.1		7.43	2.43	
	Three proteins		2.63–4.27	2.42–4.92	
	Pavir.Db00310.1			0.42	
	Pavir.Bb00799.1	Osmotin34		6.15	
	Pavir.Bb03197.1		6.63	7.83	6.08
	Pavir.Ea03024.1			6.25	
	Pavir.Eb03806.1		5.55	6.43	5.32
	Pavir.J40731.1	Galactinol synthase 1 (biosynthesis of raffinose osmoprotectants)	3.77		
Oxidative stress	Pavir.Aa01787.1; Pavir.Aa03533.1; Pavir.Ab00019.1; Pavir.Ia03296.1	Disulfide isomerases			0.45–0.48
	Pavir.Fb02222.1		2.39	3.098	
	Pavir.Ca00381.1	Alkenal reductase	3.02		2.56
	Pavir.Ea00295.1	2-oxoglutarate (2OG) and Fe(II)-DAPendent oxygenase superfamily protein	4.11	3.99	3.16
	Pavir.Ea01261.1	Peroxidase	3.01	3.20	3.98
	Pavir.Eb03999.1		3.27	2.29	2.37
	Pavir.Ia02811.1		4.41	2.97	2.83
	Pavir.J14586.1		2.65	2.94	2.20
	Pavir.J15786.1		5.24	3.53	3.55
	Pavir.J14586.1	Calatase 1	2.65	2.94	2.20
Detoxification	31 proteins	Glutathione S-transferase; Glutaredoxin; Glutathione S-transferase; Formate dehydrogenase; metallothionein 2A; Heavy metal transport/detoxification superfamily protein; Glyoxalases	2.63–16.26	2.23–4.19	1.98–6.55
Transcriptional Regulation (TFs)	Pavir.Ia00088.1; Pavir.Ba01216.1; Pavir.Ba01215.1	Ribonucleases (post-transcriptional regulation)	2.34–7.66		
	Pavir.Fb01805.1; Pavir.Gb01520.1; Pavir.Hb00704.1; Pavir.Fa00854.1	General regulatory factor (TF (fatty acid synthase genes)			0.29–0.50
	Pavir.Ab02229.1; Pavir.Gb01377.1; Pavir.Bb02517.1	Basic helix-loop-helix (bHLH) DNA-binding superfamily protein	0.17–0.37		
	Pavir.Ib01088.1	Homeodomain-like superfamily protein	3.05		
	Pavir.Da01122.1			0.44	
	Pavir.Eb00895.1	NmrA-like negative transcriptional regulator family protein	2.92		
	Pavir.Db01330.1	PHD finger protein-related	0.37		
	Pavir.Ib01055.1	RAD-like 1	0.25		
	Pavir.J31192.1	RAD-like 1		0.24	
	Pavir.Fb01946.1	Squamosa promoter binding protein-like 9	0.31		
	Pavir.Ib04454.1	Winged-helix DNA-binding transcription factor family protein	0.29		
	Pavir.J31192.1			0.42	
	Pavir.Bb02517.1	Basic helix-loop-helix (bHLH) DNA-binding superfamily protein		0.43	
	Pavir.J25625.1	Basic-leucine zipper (bZIP) transcription factor family protein		0.40	
	Pavir.Eb00253.1	CCCH-type zinc finger family protein		0.43	
	Pavir.J10403.1	Zinc finger (C3HC4-type RING finger) family protein		0.27	0.49
	Pavir.Cb00329.1	Zinc finger (C3HC4-type RING finger) family protein			0.43
	Pavir.Ab00438.1	Zinc finger C-x8-C-x5-C-x3-H type family protein			0.46
	Pavir.Da01847.1	Growth-regulating factor 5 (Transcription activator that plays a role in the regulation of cell expansion)	0.31		
	Pavir.Fb01387.1	Multiprotein bridging factor 1A (Transcriptional coactivator)		2.83	2.54
	Pavir.Ea03968.1	Nuclear factor Y, subunit B5 (TF)		0.45	
	Pavir.Ia04404.1	Transcription activator-related	9.73	8.95	
	Pavir.Cb02016.1	Transcription elongation factor (TFIIS) family protein		0.44	
	Pavir.Ea00508.1				0.48
	Pavir.Ia02581.1	Transcription factor TFIIE, alpha subunit			0.50
	Pavir.J35507.1	Splicing factor-related	0.35		
Protein translation	2 proteins in SDT, 7 in D1W, 8 in D3W	Ribosomal subunits	0.37–0.38	0.28–0.43	0.35–0.50
	Pavir.J20711.1; Pavir.Gb01843.1			2.34–2.79	
	Pavir.J16816.1; Pavir.Ga01591.1	Eukaryotic translation initiation factors	0.35–0.38		
	Pavir.J32559.1	Nuclear transport factor 2 (NTF2) family protein with RNA binding (RRM-RBD-RNP motifs) domain		0.42	0.41
	14 proteins	Ekaryotic release factor 1-3, eukaryotic translation initiation factors; Translation elongation factor EF1B; Translation initiation factor IF2/IF5; GTP binding Elongation factor Tu family protein; Essential protein Yae1			0.40–0.50
Amino acids	Pavir.Fa00915.1	Glutamate decarboxylase (GABA)	4.80	3.69	
	Pavir.Cb00192.1	3-methylcrotonyl-CoA carboxylase 1 (MCCA) (leucine catabolism)	2.67		
	Pavir.Da02439.1	Homogentisate 1,2-dioxygenase (break down tyrosine and phenylalanine	3.03	2.84	
	Pavir.Db00968.1	Dihydrodipicolinate synthase 1(lysine biosynthesis		2.62	
	Pavir.Eb03924.1	Pyrroline-5-carboxylate (P5C) reductase (arginine and proline metabolism)			0.51
Carbohydrates Sucrose/Starch	Pavir.Ab00047.1	Sucrose invertase	0.34		
	Pavir.J01866.1; Pavir.Ia03415.1	Sucrose synthase	3.37	2.51–3.66	2.42–2.58
	Pavir.J01199.1	Starch branching enzyme 2.2	3.56		
	Pavir.Ba00222.1	Melibiase family protein	6.02	4.41	2.93
Glycolysis	Pavir.Cb01993.1	Fructose-bisphosphate aldolase 2	2.86		
	Pavir.J07576.1	Phosphoglycerate kinase	2.71	2.74	2.00
	Pavir.J15479.1	Phosphofructokinase 2		3.67	
	Pavir.J13866.1	Phosphoglycerate kinase		2.31	
	Pavir.Ib01540.1	Phosphoglycerate mutase		5.40	
	Pavir.Ga01407.1	phosphofructokinase 2			2.57
Pentose shunt	Pavir.Gb02335.1	Transketolase	2.78		
	Pavir.Db00351.1	Aldolase-type TIM barrel family protein		2.23	
TCA	Pavir.Bb02852.1	Aconitase 3	4.72		
	Pavir.J02207.1; Pavir.J15849.1	Pyruvate orthophosphate dikinase	0.15–0.23		
	Pavir.Gb01988.1 Pavir.J33969.1	Phosphoenolpyruvate carboxylase		0.43–0.41	

^a^ Proteins with a fold change value (from drought-treated to non-treated control groups) greater than two standard deviations at adjusted *p* < 0.05 and identified with a minimum of two unique peptides. ^b^ Protein accession in switchgrass annotated database Pvirgatum_v1. 1. ^c^ Protein fold change from drought treated to non-drought treated groups. ^d^ Proteins from the severe drought-treated experiment. ^e^ Proteins from the 1-day re-watering experiment. ^f^ Proteins from the 3-day re-watering experiment.

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
