# Peer review of "Comparative Proteomics of Root Apex and Root Elongation Zones Provides Insights into Molecular Mechanisms for Drought Stress and Recovery Adjustment in Switchgrass"

_proteomes, 2020, doi:10.3390/proteomes8010003_

Round 1

Reviewer 1 Report

In this paper, Ye et al presented differentially expressed proteins in Switchgrass following gradual drought stress and their drought recovery afterwards. They applied tandem mass tag (TMT) based quantitative proteomics approach and reported thousands of protein identification and hundreds of those proteins differentially expressed. Overall, manuscript is well written and experiment is well executed. However, I have several major and minor points that can be addressed to make paper easy to read and follow the content.

Peptide fractionation: Line 203: “The 48 fractions were concatenated to yield 22 samples as follows: samples 1-4 and 45-48 were combined to yield two 2nd dimension fractions; then for the remaining samples (5-44), every 20th fraction was combined.” This is not clear, how labeled peptides were pooled and how many samples were run into mass spec to report these results. Please explain them in a simple way so that non-expert readers can understand.

Mascot was used to perform data analysis. Detail mass spec parameters are needed in the method section. Also, authors report >10,000 proteins with two or more unique peptides. For someone with proteomics experience, this looks surprisingly very high numbers. It is not clear whether data were filtered at 1% FDR after Mascot search using decoy database option. If FDR was applied for Mascot result, it has to be mentioned in the method.

It is critical that authors mention whether these are individual proteins or protein groups. Mascot and other search tools report protein IDs as protein group or families as many proteins have close sequences and in many instances will not have sufficient unique sequence information (unique peptides) to report individual proteins. It is important that authors provided information how they analyzed protein isoform as it would be difficult to identify two unique peptides for many isoform. I could not find information about how protein isoforms were handled during data analysis. One good example is Plant thionin authors mentioned in table 2. There are Pavir.Ib02577.1, Hbo1571.1, ………. Does each of these protein has 2 unique peptides? Or, they are part of the same protein group. If they are counted as 3 separate proteins identified, then this is not correct and its over estimation. It’s better to explicitly describe these issues.

While I will leave the final decision to the editor, I suggest table 2 and 3 can be used as supplementary files (suggestion only).

Please make the Venn diagram easy to understand. The a, b, c, d,…. In the figure make it confusing. OR, presenting it in a tabulated form could be helpful.

Figure 4 is so busy and very hard to follow. Authors have to find a better way to present this data. One suggestion could be to find top 10 most responsive (up- and down-) functional categories for each treatment and present them side by side (again only a suggestion). I will leave this for the authors to decide but it’s really hard for me to follow this figure.

In the same figure, I am not sure why almost all pathways will be down in “B” (after 1 day of re-watering following drought) while majority will be up in 4E. I did not find any explanation for this figures except they mentioned only 12% of the proteome was up-regulated in RT2 (Line 307). They didn’t mention anything about all the downregulation. This is hard to avoid and contrast between Fig4B and 4E should be explained in details.

In the result section, authors mostly focused on providing individual protein information for individual pathway. While I am aware that this is not uncommon for this type of studies, it would have been interesting if authors compare and discuss different pathways that were impacted by the gradual drought and its recovery by re-watering. I also see authors have published a similar paper recently (Ye et al, 2016, Int. J. Mol Sci). It’s not clear to me the novelty of this manuscript from this published work. Some insights would be helpful for the reviewers and ultimately to the readers.

Addressing these issues will likely improve the quality and readability of the manuscript.

Author Response

In this paper, Ye et al presented differentially expressed proteins in Switchgrass following gradual drought stress and their drought recovery afterwards. They applied tandem mass tag (TMT) based quantitative proteomics approach and reported thousands of protein identification and hundreds of those proteins differentially expressed. Overall, manuscript is well written and experiment is well executed. However, I have several major and minor points that can be addressed to make paper easy to read and follow the content.

Peptide fractionation: Line 203: “The 48 fractions were concatenated to yield 22 samples as follows: samples 1-4 and 45-48 were combined to yield two 2nd dimension fractions; then for the remaining samples (5-44), every 20th fraction was combined.” This is not clear, how labeled peptides were pooled and how many samples were run into mass spec to report these results. Please explain them in a simple way so that non-expert readers can understand.

Changes: The 48 fractions were concatenated to yield 22 samples as follows: fractions 1-4 and 45-48 were combined to yield two samples;  then for the remaining fractions (5-44), every 20th fractions were combined ( for instance, 5 and 25; 6 and 26, 7 and 27, 8 and 28, 9 and 29, following the same order) into 20 samples. Samples were dried down at reduced pressure using a CentiVac Concentrator as described above [28-30].  Line 202-206

Mascot was used to perform data analysis. Detail mass spec parameters are needed in the method section. Also, authors report >10,000 proteins with two or more unique peptides. For someone with proteomics experience, this looks surprisingly very high numbers. It is not clear whether data were filtered at 1% FDR after Mascot search using decoy database option. If FDR was applied for Mascot result, it has to be mentioned in the method.

Mascot Daemon (Version 2.3.2; Matrix Science, Boston, MA) was used to combine .pkl files generated from the 22 fractions associated with each pooled TMT labeled sample and to query them against switchgrass annotated database (http://www.phytozome.net/search.php?show=t ext&org=Org_Pvirgatum_v1. 1). Furthermore, an exclusion list including the commonly observed peptides of keratin, porcine trypsin, and Con-A was used to avoid accumulating spectra of uninformative ions. Database search criteria were: precursor mass tolerance set to 0.05 Da and fragment tolerance set to 0.1 Da. One missed tryptic cleavage was allowed. The MS/MS data were searched with S-methylation of cysteine as a fixed modification and the oxidation of methionine residues and the deamidation of asparagine and glutamine as variable modifications. All peptide matches reported herein have an E-value < 0.05. Protein identification required at least one peptide match with an E-value < 0.05. A false discovery rate (0.05) was calculated by searching the MS/MS data using a reversed decoy database. The unique peptides (1) or non-unique peptides (0) were listed in the proteomics report.   Line 228-239

It is critical that authors mention whether these are individual proteins or protein groups. Mascot and other search tools report protein IDs as protein group or families as many proteins have close sequences and in many instances will not have sufficient unique sequence information (unique peptides) to report individual proteins. It is important that authors provided information how they analyzed protein isoform as it would be difficult to identify two unique peptides for many isoform. I could not find information about how protein isoforms were handled during data analysis. One good example is Plant thionin authors mentioned in table 2. There are Pavir.Ib02577.1, Hbo1571.1, ………. Does each of these protein has 2 unique peptides? Or, they are part of the same protein group. If they are counted as 3 separate proteins identified, then this is not correct and its over estimation. It’s better to explicitly describe these issues.

All the quantified proteins have 2 or more unique peptides.

While I will leave the final decision to the editor, I suggest table 2 and 3 can be used as supplementary files (suggestion only).

Please make the Venn diagram easy to understand. The a, b, c, d,…. In the figure make it confusing. OR, presenting it in a tabulated form could be helpful.

Figure 4 is so busy and very hard to follow. Authors have to find a better way to present this data. One suggestion could be to find top 10 most responsive (up- and down-) functional categories for each treatment and present them side by side (again only a suggestion). I will leave this for the authors to decide but it’s really hard for me to follow this figure.

In the same figure, I am not sure why almost all pathways will be down in “B” (after 1 day of re-watering following drought) while majority will be up in 4E. I did not find any explanation for this figures except they mentioned only 12% of the proteome was up-regulated in RT2 (Line 307). They didn’t mention anything about all the downregulation. This is hard to avoid and contrast between Fig4B and 4E should be explained in details.

Removed the Venn diagram, replaced with Fig.3. Functional categorization of proteins identified in root-tips of switchgrass plants under severe drought treatment (SDT), 1-day re-watering (D1W) and 3-day re-watering (D3W).  RT1: basal 1-cm root-tip (RT1) tissues; RT2: elongation/maturation region of root-tips. Functional pathways were constructed using Mapman.

Figure 4. Functional categorization of drought-induced differentially expressed proteins identified in root-tips of switchgrass plants under severe drought treatment (SDT), 1-day re-watering (D1W) and 3-day re-watering (D3W).  RT1: basal 1-cm root-tip (RT1) tissues; RT2: elongation/maturation region of root-tips.

Two supplementary tables S3, S4 were added.

In the result section, authors mostly focused on providing individual protein information for individual pathway. While I am aware that this is not uncommon for this type of studies, it would have been interesting if authors compare and discuss different pathways that were impacted by the gradual drought and its recovery by re-watering. I also see authors have published a similar paper recently (Ye et al, 2016, Int. J. Mol Sci). It’s not clear to me the novelty of this manuscript from this published work. Some insights would be helpful for the reviewers and ultimately to the readers.

Added the following paragraph, line    604-613.   A long-period of drought stress has a wide-spread impact on plants from root to the above-ground tissues. In a previous proteomics study on drought-treated leaves of switchgrass, we have identified that the ABA signaling pathway was mobilized to defend against the stress. Proteomics changes involves in global transcription and translation activities, and abundance of enzymes affecting metabolic pathways of carbohydrates, amino acids, and fatty acids also expressed significant changed under drought-treated conditons [23]. When integrated with root DEPs identified in this study, it is clear that the ABA signaling pathway was activated by drought stress but involving different proteins in leaf and roots. Similarly, proteins within the same biological process or functional pathway were found to be affected differently in roots and leaves. The identification of these individual proteins will aid in the selection of important proteins (genes) for future research to develop drought tolerant plants

Reviewer 2 Report

The author presented a study on the proteomic response of root apex and root elongation zones to drought stress and recovery adjustment in the switchgrass . Here, the authors specifically targeted their study on the root apex and root elongation zones, which provides interesting and novel information on the drought physiology. Overall, this study has clearly showed a proteomic profile for root development during drought treatment and 1 and 3 days of re-watering. The results in molecular and metabolic levels clarified the phytohormone perception and signaling, cellular activities, gene transcription and metabolic pathways in drought responses. The results are clearly presented. There are some points need to be improved.

Abstract: This section is fine.

Introduction:

This section should highlight the objectives of this study.

L85 delete “)”

Materials and Methods:

In this section the details of experimental setup and methods is described; there are some errors:

L104 delete “.”

L106 delete “.”

L126 “cubes” changes to “tubes”

Results:

L286 “STD” changes to “SDT”

L327-342 In this paragraph, “W1D” changes to “D1W”, “W3D” changes to “D3W”

L384-L398 In this paragraph, “W1D” changes to “D1W”, “W3D” changes to “D3W”

L393 added “fold” before “in SDT”

L401-432 in this section, “STD” changes to “SDT”, “W1D” changes to “D1W”, “W3D” changes to “D3W”.

L433 added “)” after “DEPs”

L440-480 in this section, “STD” changes to “SDT”, “W1D” changes to “D1W”, “W3D” changes to “D3W”.

L443 “;” changes to “:”

L453 delete “cotransporter”

L455 please rephrase this sentence “Several glycolysis pathway proteins”

Discussion:

In this section, “STD” should be changed to “SDT”, “W1D” changes to “D1W”, “W3D” changes to “D3W”.

L526 “Expansins” changes to “expansins”, “Laccase” changes to “laccase”

L527-528 delete the underline

L530 delete the underline

Reference:

The formation should be carefully rechecked according to the latest format requirements.

Other comments: 1) for figure 4, please add another figure (G), which can represent the number of proteins (up and down) in some of more variable ways in the RT1 and RT2. Moreover, figure G also shows that the number of DEPs in some of more variable ways between different treatments (SDT, D1W, D3W) and different root sections (RT1 and RT2). It helps the reader to understand. 2) in table 2 and table 3, some protein accessions are not presented, in my opinion, the protein accessions should be listed in the supplementary materials, such as in table 2, the cell wall biological process. 3) the authors should choose a few DEPs for verification. 4) the picture of protein electrophoresis (SDS-PAGE) should be provied in the supplementary material.

Author Response

Comments and Suggestions for Authors

The author presented a study on the proteomic response of root apex and root elongation zones to drought stress and recovery adjustment in the switchgrass . Here, the authors specifically targeted their study on the root apex and root elongation zones, which provides interesting and novel information on the drought physiology. Overall, this study has clearly showed a proteomic profile for root development during drought treatment and 1 and 3 days of re-watering. The results in molecular and metabolic levels clarified the phytohormone perception and signaling, cellular activities, gene transcription and metabolic pathways in drought responses. The results are clearly presented. There are some points need to be improved.

Abstract: This section is fine.

Introduction:

This section should highlight the objectives of this study.

Added:  The root tip has three main zones: a zone of cell division (cells are actively dividing), a zone of elongation (cells increase in length), and a zone of maturation. Due to these distinct biological processes, the drought-induced genes/proteins in these separate zones should vary.  In this study, we have conducted analysis of drought-induced proteomic changes in the basal cell division and upper elongation/maturation zones in switchgrass during drought treatment and recovery processes. The objective is to identify proteins that confer cell-specific or universal tolerance mechanisms against drought stress .

Line  97-103. 

L85 delete “)”, could not find it.

Materials and Methods:

In this section the details of experimental setup and methods is described; there are some errors:

L104 delete “.”; changed

L106 delete “.”Changed.

L126 “cubes” changes to “tubes” 

Results:

L286 “STD” changes to “SDT”, Changed.

L327-342 In this paragraph, “W1D” changes to “D1W”, “W3D” changes to “D3W”, Changed.

L384-L398 In this paragraph, “W1D” changes to “D1W”, “W3D” changes to “D3W”, Changed.

L393 added “fold” before “in SDT”, Changed.

L401-432 in this section, “STD” changes to “SDT”, “W1D” changes to “D1W”, “W3D” changes to “D3W”.

L433 added “)” after “DEPs”: changed. 

L440-480 in this section, “STD” changes to “SDT”, “W1D” changes to “D1W”, “W3D” changes to “D3W”.

Changed.

L443 “;” changes to “:”: changed.  

L453 delete “cotransporter”; deleted.

L455 please rephrase this sentence “Several glycolysis pathway proteins”

 Several proteins in glycolysis pathway, line 499

Discussion:

In this section, “STD” should be changed to “SDT”, “W1D” changes to “D1W”, “W3D” changes to “D3W”.

L526 “Expansins” changes to “expansins”, “Laccase” changes to “laccase”

Line 570: Changed

L527-528 delete the underline

L530 delete the underline

 Line 571-572: Changed

Reference:

The formation should be carefully rechecked according to the latest format requirements.

Other comments: 1) for figure 4, please add another figure (G), which can represent the number of proteins (up and down) in some of more variable ways in the RT1 and RT2. Moreover, figure G also shows that the number of DEPs in some of more variable ways between different treatments (SDT, D1W, D3W) and different root sections (RT1 and RT2). It helps the reader to understand. 2) in table 2 and table 3, some protein accessions are not presented, in my opinion, the protein accessions should be listed in the supplementary materials, such as in table 2, the cell wall biological process. 3) the authors should choose a few DEPs for verification. 4) the picture of protein electrophoresis (SDS-PAGE) should be provided in the supplementary material. 

Fig. 4 provided the information of the number of up- and dn-regulated proteins in different functional groups. 

Picture of protein SDS-PAGE gel.   The experiment was part of dissertation for Zhujia Ye six years ago.  It has been a standard protocol in my lab to run SDS-PAGE gel for all proteomics experiments. Unfortunately,  I could not locate the gel picture for this experiment in my file.  Zhujia conducted all the experiment very diligently. I am sorry I could not provide this information at this time.